# Multi-UAV Clustered NOMA for Covert Communications: Joint Resource Allocation and Trajectory Optimization

**Xiaofei Qin, Xu Wu \*, Mudi Xiong \*, Ye Liu and Yue Zhang**

School of Information Science and Technology, Dalian Maritime University, Dalian 116026, China
\* Correspondence: xuwu@dlmu.edu.cn (X.W.); xiongmudi@dlmu.edu.cn (M.X.); Tel.: +86-15698856708 (X.W.)

**Abstract:** Due to strong survivability and flexible scheduling, multi-UAV (Unmanned Aerial Vehicle)-assisted communication networks have been widely used in civil and military fields. However, the open accessibility of wireless channels brings a huge risk of privacy disclosure to UAV-based networks. This paper considers a multi-UAV-assisted covert communication system based on Wireless Powered Communication (WPC) and Clustered-Non-Orthogonal-Multiple-Access (C-NOMA), aiming to hide the transmission behavior between UAVs and legitimate ground users (LGUs). Specifically, the UAVs serve as aerial base stations to provide services to LGUs, while avoiding detection by the ground warden. In order to improve the considered covert communication performance, the average uplink covert rate of all clusters in each slot is maximized by jointly optimizing the cluster scheduling variable, subslot allocation, LGU transmit power and multi-UAV trajectory subject to covertness constraints. The original problem is a mixed integer non-convex problem, which are typically difficult to solve directly. To solve this challenge, this paper decouples it into four sub-problems and solves the sub-problems by alternating iterations until the objective function converges. The simulation results show that the proposed multi-UAV-assisted covert communication scheme can effectively improve the average uplink covert rate of all clusters compared with the benchmark schemes.

**Keywords:** multi-UAV; covert communication; Clustered-Non-Orthogonal-Multiple-Access; subslot allocation





## 1. Introduction

In recent years, unmanned aerial vehicle (UAV)-assisted wireless communication has attracted great attention from civil and military applications, such as disaster rescue, monitoring and data relaying [1–3]. Benefiting from high altitude, UAVs can establish Line-of-Sight (LoS) links for air–ground communication, which provides a significant performance gain compared to the traditional cellular communication with typically Non-Line-of-Sight (NLoS) transmission.

In particular, when the user device is located in the disaster area, the base station (BS) may be damaged and, thus, cannot provide reliable services to the user. Due to flexible deployment and high mobility, UAVs are able to act as flying base stations and can provide wireless connections to disconnected user devices [4]. In addition, UAVs can also collect mobile data and disseminate information to assist with wireless services of users with versatile requirements [5].

Although UAVs have myriad advantages, the energy of user devices is limited [6]. Therefore, there is an urgent problem to solve in UAV-assisted wireless communication regarding how to simultaneously provide a convenient energy supply and information access services for user devices. In recent years, Wireless Powered Communication (WPC) has developed into a solution for energy supply in wireless communication [7,8].

WPC uses radio frequency (RF) signals, whose energy can be partially collected, converted and stored in user devices [9]. The user device can harvest energy from the downlink RF signal and then send information to the uplink by utilizing the harvested

energy. Cheng et al. [10] studied a UAV-assisted wireless powered Internet of Things. Specifically, when hovering in the air, the UAV communicates and supplies power to user devices, thereby, effectively dealing with the energy limitations of user devices and prolonging their endurance.

On the other hand, since the amount of user devices has recently experienced exponential growth, it is crucial to utilize spectrum resources more efficiently. Traditional orthogonal multiple access (OMA) has difficulty meeting the requirements of large-scale connections of user devices [11]. Fortunately, Non-Orthogonal-Multiple-Access (NOMA) can adapt to the large-scale connectivity and high-efficiency spectrum utilization in wireless communication networks [12–16]. Compared with OMA, NOMA provides services to multiple user devices over the same time, frequency or pattern by leveraging power multiplexing that has higher throughput, lower delay communication and improved spectral efficiency [17–22].

In addition, NOMA recovers the information from the receiver using successive interference cancellation (SIC) technology. However, the complexity brought by SIC increases linearly with increases in the number of user devices, resulting in significant increases in processing delays [23]. Therefore, when serving as the flying base station, the UAV can leverage clustered NOMA (C-NOMA), which can not only effectively improve spectrum resource utilization but also reduce the decoding difficulty at the receiver to thereby achieve a good trade-off between high throughput and low decoding complexity [24].

Although UAV-assisted wireless communication has brought technical conveniences, new challenges have also followed. The dominant LoS link makes the confidential information transmitted by the UAV more easily intercepted by the warden on the ground due to the broadcast nature of the wireless channel in the UAV network. Some works have studied such security issues from the perspective of Physical Layer Security (PLS) [25–27]. Though PLS can protect the information content of wireless communications, it is highly necessary to hide legal communication behaviors in confidential or private scenarios, such as distributed ground reconnaissance systems.

Fortunately, covert communication technology can meet the requirements of hiding UAV information transmissions. In covert communication, the sender transmits information to the receiver in a covert manner to prevent the warden from detecting the wireless transmissions [28,29]. Bash et al. [30] proved that transmissions of exceeding $\mathcal{O}(\sqrt{n})$ bits of information will result in the warden's detection error probability approaching zero. Goeckel et al. [31] showed that, when the warden cannot know the noise power accurately, the sender can covertly transmit $\mathcal{O}(\sqrt{n})$ bits of information to the receiver.

Yan et al. [32] analyzed the covert performance of air–ground wireless communication systems and then jointly optimized the transmission power and altitude of UAVs. Zhou et al. [33] assumed that the warden had an unlimited number of observations in each time-slot, and synergistically optimized the transmission power and trajectory of UAVs to maximize the covert throughput of the air–ground communication system. Wang et al. [34] improved the communication quality between the UAV and legitimate ground users (LGUs) outside the monitoring area to the greatest extent by jointly optimizing the transmission power and fixed position of the UAV. Jiang et al. [35] resolved the problem of robust resource allocation and UAV trajectory optimization to maximize the average covert rate of multiple LGUs.

From the above analysis, we can see that there are two shortcomings in existing research on UAV covert communication: (1) The single UAV or single user situation has been primarily studied, while multi-UAV or multi-user cases have been overlooked. (2) The influence of limited energy storage on continuous communication of user devices is not considered. As far as we know, there has been no technical attention to the orchestration of WPC and NOMA for multi-UAV-assisted covert communication. Therefore, this paper proposes a multi-UAV-assisted covert communication system, where WPC is utilized to provide wireless radio frequency energy to LGUs while C-NOMA is exploited to improve the system spectral efficiency. The contributions of this paper are summarized as follows.

- A multi-UAV-assisted covert communication system based on WPC and C-NOMA is proposed. First, a traversal search strategy is adopted to dynamically cluster LGUs. NOMA is applied to intra-cluster LGUs, while TDMA is adopted by inter-clusters. For multi-user scenarios, we deploy multiple UAVs to serve LGUs. In the downlink sub-slot, LGUs collectively harvest energy from UAV wireless signals. In the uplink sub-slot, LGUs transmit data to UAVs while the warden attempts to detect data transmission between UAVs and LGUs with a limited number of observations.
- The closed-form analytic expression of the minimum average detection error probability of the warden is derived. Specifically, when the warden can select the detection threshold, the expressions of false-alarm and missed-detection probability are derived, and then the detection error probability at the warden is determined. By analyzing the detection performance, the covertness constraint is determined. According to periodic flight mode, the covert communication performance of the proposed system is investigated.
- A joint optimization scheme for multi-UAV trajectory and resource allocation is proposed. Since there are co-channel interferences between UAVs and LGUs in the uplink, the associated scheduling between UAVs and LGUs should be carefully designed to select the appropriate LGU to upload data to the specific UAV. The objective of this scheme is to maximize the average uplink covert rate of all clusters over the entire flight cycle by reasonably designing the cluster scheduling variable, sub-slot allocation, LGU transmit power and multi-UAV trajectory under the premise of the covertness constraints.
- A high-efficiency algorithm is proposed. In order to enhance the performance of the covert communication system, an alternative iterative algorithm is proposed with respect to the cluster scheduling variable, sub-slot allocation, multi-UAV transmission power and trajectory. Specifically, the originally non-convex optimization problem is divided into four sub-problems. Then, each non-convex sub-problem is transformed into a convex one by variable relaxation and the successive convex approximation method and is then solved using the CVX tool. Finally, the optimization objective converges through alternate iterations.

The organizational structure of this paper is as follows. In Section 2, the multi-UAV-assisted covert communication system model is given, and then the corresponding optimization problem is formulated. In Section 3, the proposed algorithm and its derivations are presented. Section 4 gives the simulation parameters, numerical results and necessary discussions. Finally, Section 5 concludes this paper.

## 2. System Model and Problem Formulation

As shown in Figure 1, this paper considers a multi-UAV-assisted covert communication system based on WPC and C-NOMA, consisting of $M$ UAVs, $G$ legitimate ground users (LGUs) and a ground warden, $M, G \in \mathbf{N}^+$. In Figure 1, $G$ LGUs are divided into $J$ clusters, and each cluster contains $K$ LGUs, $G = JK, J, K \in \mathbf{N}^+$, and $j \in \mathcal{J} = \{1, 2, ..., J\}, k \in \mathcal{K} = \{1, 2, ..., K\}$. The $k$-th LGU in the $j$-th cluster is denoted as $G_{j,k}$, whose horizontal coordinate is expressed as $\mathbf{W}_{j,k} = \left[ x_{j,k}, y_{j,k} \right]^{\mathrm{T}}$.

The UAVs transmit energy to LGUs, and LGUs transmit confidential information to UAVs, while the warden detects the communication behavior between UAVs and LGUs. Assume that each UAV flies at a fixed altitude $H$ above the horizontal ground and has the same flight cycle $T$. Without losing generality, in a 3D Cartesian coordinate system, the horizontal coordinate of UAVs $m \in \mathcal{M} = \{1, 2, ..., M\}$ in timeslot $l \in \mathcal{L} = \{1, 2, ..., L\}$ is denoted as $\mathbf{q}_m[l] = [x_m[l], y_m[l]]^{\mathrm{T}}$, and the warden's position is expressed as $\mathbf{q}_{\mathrm{w}} = [x_{\mathrm{w}}, y_{\mathrm{w}}]^{\mathrm{T}}$.

When all UAVs fly in periodic mode, the terminal position $\mathbf{q}_m[L]$ coincides with the starting position $\mathbf{q}_m[1]$. Suppose that the UAVs have a constant speed in each timeslot, and

the maximum speed of UAVs is denoted as $V_{\max}$. Therefore, the mobility constraints of UAVs can be written as follows.

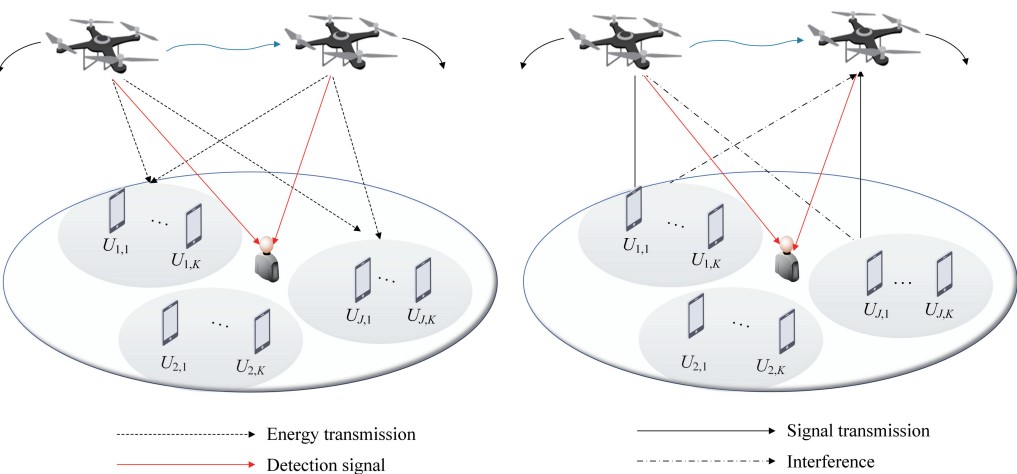

**Figure 1.** Multi-UAV-assisted covert communication system.

$$\mathbf{q}_m[1] = \mathbf{q}_m[L], \forall m \tag{1}$$

$$||\mathbf{q}_m[l+1] - \mathbf{q}_m[l]||^2 \leq \left(\frac{V_{\max}T}{L}\right)^2, l = 1, 2, ..., L-1 \tag{2}$$

$$||\mathbf{q}_m[l] - \mathbf{q}_c[l]||^2 \geq d_{\min}^2, \forall m, l, m \neq c \tag{3}$$

where $d_{\min}$ stands for the minimum safe distance between UAVs. Each timeslot is partitioned into downlink sub-slot $\delta_0[l]$ for energy transfer and uplink sub-slot $\delta_1[l]$ for data transmission, respectively. As the air–ground wireless channel is dominated by the LoS link, the channel gains between the $m$-th UAV and the $k$-th LGU in the $j$-th cluster, and between the $k$-th LGU in the $j$-th cluster and the warden can be, respectively, represented as

$$h_{j,k}[l] = \frac{\beta_0}{d_{j,k}^2[l]}, \forall l, j, k \tag{4}$$

$$h_{\mathrm{gw}}[l] = \frac{\beta_0}{d_{\mathrm{gw}}^2[l]}, \forall l \tag{5}$$

where $\beta_0$ stands for the channel power gain at the unit distance $d_0 = 1m$. $d_{j,k}[l]$ represents the distance between the $k$-th LGU in the $j$-th cluster and UAVs in the $l$-th timeslot, and $d_{\mathrm{gw}}[l]$ represents the distance between the $k$-th LGU in the $j$-th cluster and the warden in the $l$-th timeslot, which are, respectively, written as

$$d_{j,k}[l] = \sqrt{H^2 + \left\|\mathbf{q}_m[l] - \mathbf{W}_{j,k}\right\|^2}, \forall l, j, k \tag{6}$$

$$d_{\mathrm{gw}}[l] = \sqrt{H^2 + \left\|\mathbf{W}_{j,k} - \mathbf{q}_{\mathrm{w}}\right\|^2}, \forall j, k, l \tag{7}$$

In particular, the channel power gain difference of intra-cluster LGUs in the $j$-th cluster in the $l$-th timeslot can be expressed as

$$\Delta h_j[l] = \sum_{r=1}^{N-1} \left| |h_{j,r+1}[l]|^2 - |h_{j,r}[l]|^2 \right|, \forall j, l \tag{8}$$

The sum of the channel power gains of all clusters in the $l$-th timeslot is given by

$$S[l] = \sum_{j=1}^{J} \Delta h_j[l], \forall l \tag{9}$$

In the downlink, the energy harvested by the $k$-th LGU in the $j$-th cluster in the $l$-th timeslot LGUs can be written as

$$E_{j,k}[l] = \mu h_{j,k}[l] P_m[l] \delta_0[l], \forall l, j, k, m \tag{10}$$

where $\mu$ is the energy conversion efficiency, $P_m[l]$ denotes the transmit power of the $m$-th UAV in the $l$-th timeslot, satisfying the following constraint

$$0 \leq P_m[l] \leq P_{\text{max}}, \forall l \tag{11}$$

In the uplink, LGUs upload data to UAVs by utilizing the harvested energy. The transmit power of the $k$-th LGU in the $j$-th cluster in the $l$-th timeslot is given by

$$P_{j,k}[l] = \frac{E_{j,k}[l]}{\delta_1[l]} = \frac{\mu h_{j,k}[l] P_m[l] \delta_0[l]}{\delta_1[l]}, \forall l, j, k, m \tag{12}$$

Assume that each UAV only communicates with an associated cluster in each timeslot. Define a binary UAV-cluster scheduling variable $\alpha_{j,m}[l]$. If the $j$-th cluster uploads data to the $m$-th UAV in sub-slot $\delta_1[l]$, then $\alpha_{j,m}[l] = 1$; otherwise, $\alpha_{j,m}[l] = 0$. Hence, the scheduling variable is subject to the following constraints

$$\alpha_{j,m}[l] \in \{0, 1\}, \forall j, m, l \tag{13}$$

$$\sum_{j=1}^{J} \alpha_{j,m}[l] \leq 1, \forall m, l \tag{14}$$

$$\sum_{m=1}^{M} \alpha_{j,m}[l] \leq 1, \forall j, l \tag{15}$$

In the $l$-th timeslot, the uplink covert rate of the $k$-th LGU $G_{j,k}$ in the $j$-th cluster can be represented as

$$R_{j,k}[l] = \sum_{m=1}^{M} \log_2 \left( 1 + \frac{P_{j,k}[l] h_{j,k}[l]}{\sum_{b=k+1}^{K} P_{j,b}[l] h_{j,b}[l] + \sum_{c=1,c \neq m}^{M} \sum_{f=1}^{J} \sum_{k=1}^{K} \alpha_{c,f}[l] P_{f,k}[l] h_{f,k}[l] + N_0} \right), \forall l \tag{16}$$

where $N_0$ stands for the noise power. $\sum_{b=k+1}^{K} P_{j,b}[l] h_{j,b}[l]$ represents the intra-cluster interference, $\sum_{c=1,c \neq m}^{M} \sum_{f=1}^{J} \sum_{k=1}^{K} \alpha_{c,f}[l] P_{f,k}[l] h_{f,k}[l]$ is the co-channel interference generated by UAV $c(c = 1, ..., M, c \neq m)$ to the $j$-th cluster when UAV $m$ serves the $j$-th cluster in the $l$-th time slot. Therefore, the average uplink covert rate of the $j$-th cluster over the entire flight cycle of the UAVs can be written as

$$R_j[l] = \frac{1}{L} \sum_{l=1}^{L} \sum_{k=1}^{K} R_{j,k}[l], \forall l, j \tag{17}$$

In covert communication, we suppose that the warden makes a limited number of observations and then determines whether there is data transmission between the UAVs and LGUs based on the received signals, i.e.,

$$y_{\text{w}}^{(i)}[l] = \begin{cases} n_{\text{w}}[l], & \mathcal{H}_0 \\ \sqrt{\frac{\beta_0 P_{j,k}[l]}{H^2 + \|\mathbf{W}_{j,k} - \mathbf{q}_{\text{w}}\|^2}} x[l] + n_{\text{w}}[l], & \mathcal{H}_1 \end{cases} \tag{18}$$

where the null hypothesis $\mathcal{H}_0$ means that there is no data transmission between UAVs and LGUs, while the alternative hypothesis $\mathcal{H}_1$ indicates that LGUs upload data. $y_{\text{w}}^{(i)}[l]$ represents the signal received by the warden during the $i$-th detection in the $l$-th timeslot, $i = 1, 2, ..., I$, $x[l]$ stands for the transmitted signal of UAVs in the $l$-th timeslot, following the complex Gaussian distribution with mean value of 0 and variance of 1, and $n_{\text{w}}[l]$ represents the additive white Gaussian noise (AWGN) at the warden in the $l$-th timeslot. As such, the signal received by the warden is subject to

$$y_{\text{w}}^{(i)}[l] \sim \begin{cases} CN(0, \sigma^2), & \mathcal{H}_0 \\ CN\left(0, P_{j,k}[l]h_{\text{gw}}[l] + \sigma^2\right), & \mathcal{H}_1 \end{cases} \tag{19}$$

where $\sigma^2$ is the power of AWGN.

In practice, the warden may not accurately know the power of AWGN due to the dynamic nature of communication environments. Therefore, following [36], the noise uncertainty is considered in this paper. Noise power $\sigma_w^2$ is a random variable subject to a known distribution. We assume that $\sigma_{w,\text{dB}}^2 \in [\widehat{\sigma}_{\text{dB}}^2 - \rho_{\text{dB}}, \widehat{\sigma}_{\text{dB}}^2 + \rho_{\text{dB}}]$ follows a uniform distribution in the value range. Then, the probability density function (PDF) of $\sigma_w^2$ is given by

$$f_{\sigma_w^2}(x) = \begin{cases} \frac{1}{2x \ln(\rho)}, & \text{if } \frac{1}{\rho}\widehat{\sigma}^2 \le x \le \rho\widehat{\sigma}^2, \\ 0, & \text{otherwise.} \end{cases} \tag{20}$$

where $\sigma_{w,\text{dB}}^2 = 10\log_{10}(\sigma_w^2), \widehat{\sigma}_{\text{dB}}^2 = 10\log_{10}(\widehat{\sigma}^2)$, $\widehat{\sigma}^2$ denotes the nominal noise power, $\rho_{\text{dB}} = 10\log_{10}(\rho)$ is a parameter for measuring the uncertainty of noise, and $\rho \ge 1$.

Suppose that $\mathcal{D}_0$ and $\mathcal{D}_1$, respectively, represent favorable decisions for $\mathcal{H}_0$ and $\mathcal{H}_1$. The false alarm probability and the missed detection probability in the $l$-th timeslot are written as $\mathbb{P}_{\text{FA}}[l] \triangleq \mathbb{P}(\mathcal{D}_1 | \mathcal{H}_0)$ and $\mathbb{P}_{\text{MD}}[l] \triangleq \mathbb{P}(\mathcal{D}_0 | \mathcal{H}_1)$. Define $\xi[l] = \mathbb{P}_{\text{FA}}[l] + \mathbb{P}_{\text{MD}}[l]$ as the warden's detection error probability in the $l$-th timeslot. The UAV goal is to make $\xi[l]$ in each timeslot approach 1, which can be written as

$$\xi[l] = \mathbb{P}_{\text{FA}}[l] + \mathbb{P}_{\text{MD}}[l] \ge 1 - \varepsilon, \forall l \tag{21}$$

where $0 \le \varepsilon \le 1$ stands for an arbitrarily small value that determines the covertness constraint, also known as the coverage constraint.

The objective of this paper is to simultaneously optimize the cluster scheduling variable $\mathbf{A} = \{\alpha_{j,m}[l], \forall j, m, l\}$, uplink sub-slot $\boldsymbol{\delta_1} = \{\delta_1[l], \forall l\}$, downlink sub-slot $\boldsymbol{\delta_0} = \{\delta_0[l], \forall l\}$, transmit power of LGUs $\mathbf{P} = \{p_{j,k}[l], \forall j, k, l\}$ and multi-UAV trajectory $\mathbf{Q} = \{\mathbf{q}_m[l], \forall m, l\}$ under the premise of covertness constraints, to maximize the average uplink covert rate of all clusters over the entire flight cycle. The corresponding optimization problem can be written as

$$\max_{\mathbf{A}, \boldsymbol{\delta_0}, \boldsymbol{\delta_1}, \mathbf{P}, \mathbf{Q}} \quad \frac{1}{L} \sum_{l=1}^{L} \sum_{k=1}^{K} R_{j,k}[l] \tag{22a}$$

$$\text{s.t.} \quad \sum_{k=1}^{K} R_{j,k}[l] \ge \lambda, \forall l \tag{22b}$$

$$\delta_0[l] + \delta_1[l] \le \delta, \forall l \tag{22c}$$

$$(1), (2), (3), (13), (14), (15), (21) \tag{22d}$$

## 3. Proposed Solution

In this section, for the proposed mixed integer nonconvex optimization problem (22), we first decompose it into four sub-problems with respect to the cluster scheduling variable $\mathbf{A}$, uplink and downlink sub-slots $\boldsymbol{\delta_1}$ and $\boldsymbol{\delta_0}$, LGU transmit power $\mathbf{P}$ and multi-UAV trajectory $\mathbf{Q}$. Then, the relaxation variable method and SCA technique are utilized

to transform the corresponding subproblems into convex forms. Finally, the optimization objective is converged upon through alternate iterations.

### 3.1. Subslot Optimization

Given **A**, **P** and **Q**, the optimization problem with respect to sub-slots $\delta_1$ and $\delta_0$ can be described as follows

$$\max_{\delta_0,\delta_1} \quad \frac{1}{L}\sum_{l=1}^{L}\sum_{k=1}^{K} R_{j,k}[l] \tag{23a}$$

$$\text{s.t.} \quad \sum_{k=1}^{K} R_{j,k}[l] \geq \lambda, \forall l \tag{23b}$$

$$\delta_0[l] + \delta_1[l] \leq \delta, \forall l \tag{23c}$$

First, we simplify (23a) according to the following Lemma 1. Then, a resource allocation algorithm combined bisection method and Lagrangian multiplier method is proposed to optimize the uplink and downlink sub-slots.

**Lemma 1.** *For any given multi-UAV trajectory, the uplink covert rate of the j-th cluster can be written as*

$$R_j = \frac{1}{L}\sum_{l=1}^{L}\sum_{k=1}^{K} R_{j,k}[l]$$

$$= \frac{1}{L}\sum_{l=1}^{L}\sum_{m=1}^{M} \alpha_{j,m}[l]\delta_1[l]\log_2\left(1 + \sum_{k=1}^{K}\frac{a_{j,k}[l]\delta_0[l]}{\delta_1[l]} + \sum_{c=1,c\neq m}^{M}\sum_{f=1}^{J}\sum_{k=1}^{K}\alpha_{c,f}[l]\frac{a_{f,k}[l]\delta_0[l]}{\delta_1[l]}\right) \tag{24}$$

*where $a_{j,k}[l] = \frac{\mu h_{j,k}^2[l]P_m[l]}{N_0}$.*

**Proof.** The detailed proof is presented in Appendix A.
According to Lemma 1, the objective function in (23a) can be expressed as

$$R_j = \frac{1}{L}\sum_{l=1}^{L}\sum_{k=1}^{K} R_{j,k}[l]$$

$$= \frac{1}{L}\sum_{l=1}^{L}\sum_{m=1}^{M} \alpha_{j,m}[l]\delta_1[l]\log_2\left(1 + \sum_{k=1}^{K}\frac{a_{j,k}[l]\delta_0[l]}{\delta_1[l]} + \sum_{c=1,c\neq m}^{M}\sum_{f=1}^{J}\sum_{k=1}^{K}\alpha_{c,f}[l]\frac{a_{f,k}[l]\delta_0[l]}{\delta_1[l]}\right) \tag{25}$$

Then, the second-order partial derivatives of (25) with respect to $\delta_0[l]$ and $\delta_1[l]$ are, respectively, given by

$$\frac{\partial^2 R_j}{\partial \delta_0[l]^2} =$$

$$-\frac{1}{L\ln 2}\sum_{l=1}^{L}\sum_{m=1}^{M}\alpha_{j,m}[l]\frac{\left(\sum_{k=1}^{K}a_{j,k}[l]\delta_0[l] + \sum_{c=1,c\neq m}^{M}\sum_{f=1}^{J}\sum_{k=1}^{K}\alpha_{c,f}[l]a_{f,k}[l]\delta_0[l]\right)^2}{\delta_1[l]\left(\delta_1[l] + \sum_{k=1}^{K}a_{j,k}[l]\delta_0[l] + \sum_{c=1,c\neq m}^{M}\sum_{f=1}^{J}\sum_{k=1}^{K}\alpha_{c,f}[l]a_{f,k}[l]\delta_0[l]\right)^2} < 0 \tag{26}$$

$$\frac{\partial^2 R_j}{\partial \delta_1[l]^2} =$$

$$-\frac{1}{L\ln 2}\sum_{l=1}^{L}\sum_{m=1}^{M}\alpha_{j,m}[l]\frac{\left(\sum_{k=1}^{K}a_{j,k}[l]\delta_0[l] + \sum_{c=1,c\neq m}^{M}\sum_{f=1}^{J}\sum_{k=1}^{K}\alpha_{c,f}[l]a_{f,k}[l]\delta_0[l]\right)^2}{\delta_1[l]\left(\delta_1[l] + \sum_{k=1}^{K}a_{j,k}[l]\delta_0[l] + \sum_{c=1,c\neq m}^{M}\sum_{f=1}^{J}\sum_{k=1}^{K}\alpha_{c,f}[l]a_{f,k}[l]\delta_0[l]\right)^2} < 0 \tag{27}$$

Clearly, $R_j$ in (25) is concave with respect to $\delta_0[l]$ and $\delta_1[l]$. As the constraints in (23b) and (23c) are affine functions, (23) is a convex optimization problem, which can be treated by the Lagrange multiplier method. The Lagrangian function is given as follows

$$
\begin{aligned}
\ell = {} & \frac{1}{L}\sum_{l=1}^{L}\sum_{m=1}^{M}\alpha_{j,m}[l]\delta_1[l]\log_2\left(1+\sum_{k=1}^{K}\frac{a_{j,k}[l]\delta_0[l]}{\delta_1[l]}+\sum_{c=1,c\neq m}^{M}\sum_{f=1}^{J}\sum_{k=1}^{K}\alpha_{c,f}[l]\frac{a_{f,k}[l]\delta_0[l]}{\delta_1[l]}\right) \\
& +\alpha(\delta-\delta_0[l]-\delta_1[l]) \\
& +\beta\left(\sum_{m=1}^{M}\alpha_{j,m}[l]\delta_1[l]\log_2\left(1+\sum_{k=1}^{K}\frac{a_{j,k}[l]\delta_0[l]}{\delta_1[l]}+\sum_{c=1,c\neq m}^{M}\sum_{f=1}^{J}\sum_{k=1}^{K}\alpha_{c,f}[l]\frac{a_{f,k}[l]\delta_0[l]}{\delta_1[l]}\right)-\lambda\right)
\end{aligned}
\tag{28}
$$

where $\alpha$ and $\beta$ denote Lagrange multipliers. The first-order partial derivatives of (29) with respect to $\delta_0[l]$ and $\delta_1[l]$ are, respectively, represented as

$$
\begin{aligned}
& \frac{\partial\ell}{\partial\delta_0[l]} = \\
& \frac{1}{\ln 2}\left(\frac{1}{L}+\beta\right)\sum_{m=1}^{M}\alpha_{j,m}[l]\frac{\left(\sum_{k=1}^{K}a_{j,k}[l]+\sum_{c=1,c\neq m}^{M}\sum_{f=1}^{J}\sum_{k=1}^{K}\alpha_{c,f}[l]a_{f,k}[l]\right)\delta_1[l]}{\delta_1[l]+\sum_{k=1}^{K}a_{j,k}[l]\delta_0[l]+\sum_{c=1,c\neq m}^{M}\sum_{f=1}^{J}\sum_{k=1}^{K}\alpha_{c,f}[l]a_{f,k}[l]\delta_0[l]}-\alpha
\end{aligned}
\tag{29}
$$

$$
\begin{aligned}
\frac{\partial\ell}{\partial\delta_1[l]} = {} & \left(\frac{1}{L}+\beta\right)\sum_{m=1}^{M}\alpha_{j,m}[l]\log_2\left(\left(1+\sum_{k=1}^{K}\frac{a_{j,k}[l]\delta_0[l]}{\delta_1[l]}+\sum_{c=1,c\neq m}^{M}\sum_{f=1}^{J}\sum_{k=1}^{K}\alpha_{c,f}[l]\frac{a_{f,k}[l]\delta_0[l]}{\delta_1[l]}\right)\right. \\
& \left.-\frac{1}{\ln 2}\frac{\left(\sum_{k=1}^{K}a_{j,k}[l]+\sum_{c=1,c\neq m}^{M}\sum_{f=1}^{J}\sum_{k=1}^{K}\alpha_{c,f}[l]a_{f,k}[l]\right)\delta_0[l]}{\delta_1[l]+\sum_{k=1}^{K}a_{j,k}[l]\delta_0[l]+\sum_{c=1,c\neq m}^{M}\sum_{f=1}^{J}\sum_{k=1}^{K}\alpha_{c,f}[l]a_{f,k}[l]\delta_0[l]}\right)-\alpha
\end{aligned}
\tag{30}
$$

According to the Karush–Kuhn–Tucker (KKT) conditions, the optimal solution can be obtained by setting (29) and (30) to zero. We let (29) be equal to (30), and then Equation (31) can be given by

$$
\begin{aligned}
& \sum_{m=1}^{M}\alpha_{j,m}[l]\log_2\left(1+\sum_{k=1}^{K}\frac{a_{j,k}[l]\delta_0[l]}{\delta_1[l]}+\sum_{c=1,c\neq m}^{M}\sum_{f=1}^{J}\sum_{k=1}^{K}\alpha_{c,f}[l]\frac{a_{f,k}[l]\delta_0[l]}{\delta_1[l]}\right) \\
& -\frac{1}{\ln 2}\frac{\left(\sum_{k=1}^{K}a_{j,k}[l]+\sum_{c=1,c\neq m}^{M}\sum_{f=1}^{J}\sum_{k=1}^{K}\alpha_{c,f}[l]a_{f,k}[l]\right)(\delta_0[l]+\delta_1[l])}{\delta_1[l]+\left(\sum_{k=1}^{K}a_{j,k}[l]+\sum_{c=1,c\neq m}^{M}\sum_{f=1}^{J}\sum_{k=1}^{K}\alpha_{c,f}[l]a_{f,k}[l]\delta_0[l]\right)\delta_0[l]}=0
\end{aligned}
\tag{31}
$$

According to the constraint (23c), (31) can be further expressed as (32)

$$
\begin{aligned}
& \sum_{m=1}^{M}\alpha_{j,m}[l]\log_2\left(1+\sum_{k=1}^{K}\frac{a_{j,k}[l]\delta_0[l]}{\delta-\delta_0[l]}+\sum_{c=1,c\neq m}^{M}\sum_{f=1}^{J}\sum_{k=1}^{K}\alpha_{c,f}[l]\frac{a_{f,k}[l]\delta_0[l]}{\delta-\delta_0[l]}\right) \\
& -\frac{1}{\ln 2}\frac{\left(\sum_{k=1}^{K}a_{j,k}[l]+\sum_{c=1,c\neq m}^{M}\sum_{f=1}^{J}\sum_{k=1}^{K}\alpha_{c,f}[l]a_{f,k}[l]\right)\delta}{\delta-\delta_0[l]+\left(\sum_{k=1}^{K}a_{j,k}[l]+\sum_{c=1,c\neq m}^{M}\sum_{f=1}^{J}\sum_{k=1}^{K}\alpha_{c,f}[l]a_{f,k}[l]\delta_0[l]\right)\delta_0[l]}=0
\end{aligned}
\tag{32}
$$

It is easy to find from (32) that $\delta_0[l]$ is the only variable. As such, the optimal downlink sub-slot $\delta_0^*[l]$ can be obtained by solving Equation (32). When the difference between the upper and lower thresholds of $\delta_0[l]$ reaches a certain accuracy after multiple iterations, the bisection method can be adopted to iteratively search for the downlink sub-slot $\delta_0^*[l]$. For the sake of making full use of the whole timeslot, the uplink sub-slot $\delta_1^*[l]$ can be selected as

$$\delta_1^*[l] = \delta - \delta_0^*[l] \tag{33}$$

□

### 3.2. UAV-Cluster Scheduling Optimization

Given $\delta_1$, $\delta_0$, **P** and **Q**, the optimization problem with respect to cluster scheduling variable **A** can be described as follows

$$\max_{\mathbf{A}} \quad \frac{1}{L}\sum_{l=1}^{L}\sum_{k=1}^{K} R_{j,k}[l] \tag{34a}$$

$$\text{s.t.} \quad \sum_{k=1}^{K} R_{j,k}[l] \geq \lambda, \forall l \tag{34b}$$

$$((13),(14),(15)) \tag{34c}$$

Due to the non-convexity of (34b) and binary variable **A**, (34) is also a mixed integer non-convex optimization problem. As such, we can relax $\alpha_{j,m}[l] \in \{0,1\}$ to $0 \leq \alpha_{j,m}[l] \leq 1, \forall j, m, l$ and then introduce a relaxation variable $\{\lambda_{j,k}[l]\}$. Thus, (34) can be rewritten as

$$\max_{\mathbf{A},\{\lambda_{j,k}[l]\}} \quad \frac{1}{L}\sum_{l=1}^{L}\sum_{k=1}^{K} R_{j,k}[l] \tag{35a}$$

$$\text{s.t.} \quad \sum_{k=1}^{K}\sum_{m=1}^{M} \alpha_{j,m}[l]\delta_1[l]\lambda_{j,k}[l] \geq \lambda, \forall j \tag{35b}$$

$$\log_2\left(1 + \frac{P_{j,k}[l]h_{j,k}[l]}{\sum\limits_{b=k+1}^{K} P_{j,b}[l]h_{j,b}[l] + \sum\limits_{c=1,c\neq m}^{M}\sum\limits_{f=1}^{J}\sum\limits_{k=1}^{K} \alpha_{c,f}[l]P_{f,k}[l]h_{f,k}[l] + N_0}\right) \geq \lambda_{j,k}[l], \forall j, k, m \tag{35c}$$

$$0 \leq \alpha_{j,m}[l] \leq 1, \forall j, m, l \tag{35d}$$

$$(14) \text{ and } (15) \tag{35e}$$

Clearly, the solution of (35) is also applicable to (34), because the two problems are equivalent to each other when (35c) takes the equal sign. Note that (35b) and (35c) are non-convex functions, and thus (35) is also a non-convex problem. It can be transformed into a convex optimization problem through a Taylor series expansion. Specifically, $\alpha_{j,m}[l]\lambda_{j,k}[l]$ in (35b) is first converted into the following form

$$\alpha_{j,m}[l]\lambda_{j,k}[l] = \frac{\left(\alpha_{j,m}[l] + \lambda_{j,k}[l]\right)^2}{2} - \frac{\alpha_{j,m}^2[l] + \lambda_{j,k}^2[l]}{2} \tag{36}$$

Note that $\left(\alpha_{j,m}[l] + \lambda_{j,k}[l]\right)^2$ in (36) is convex with respect to $\alpha_{j,m}[l]$ and $\lambda_{j,k}[l]$. The lower bound of $\left(\alpha_{j,m}[l] + \lambda_{j,k}[l]\right)^2$ can be obtained through Taylor series expansion as

$$\left(\alpha_{j,m}[l] + \lambda_{j,k}[l]\right)^2 \geq \theta_{j,m}^{\tilde{l}b}[l] = \left(\alpha_{j,m}^{(\tilde{l})}[l] + \lambda_{j,k}^{(\tilde{l})}[l]\right)^2 + 2\left(\alpha_{j,m}^{(\tilde{l})}[l] + \lambda_{j,k}^{(\tilde{l})}[l]\right)\left(\alpha_{j,m}[l] - \alpha_{j,m}^{(\tilde{l})}[l]\right)$$
$$+ 2\left(\alpha_{j,m}^{(\tilde{l})}[l] + \lambda_{j,k}^{(\tilde{l})}[l]\right)\left(\lambda_{j,k}[l] - \lambda_{j,k}^{(\tilde{l})}[l]\right) \tag{37}$$

Clearly, the left side of (35c) is convex with respect to $\alpha_{c,f}[l]$. Similarly, the lower bound of $R_{j,k}[l]$ can be obtained by its first-order Taylor series expansion. Thus, we have the following derivations

$$R_{j,k}[l] = \log_2\left(1 + \frac{P_{j,k}[l]h_{j,k}[l]}{\sum\limits_{b=k+1}^{K} P_{j,b}[l]h_{j,b}[l] + \sum\limits_{c=1,c\neq m}^{M}\sum\limits_{f=1}^{J}\sum\limits_{k=1}^{K}\alpha_{c,f}[l]P_{f,k}[l]h_{f,k}[l] + N_0}\right) \geq R_{j,k}^{\tilde{lb}}[l] \tag{38}$$

where

$$R_{j,k}^{\tilde{lb}}[l] = \log_2\left(1 + \frac{P_{j,k}[l]h_{j,k}[l]}{\sum\limits_{b=k+1}^{K} P_{j,b}[l]h_{j,b}[l] + \sum\limits_{c=1,c\neq m}^{M}\sum\limits_{f=1}^{J}\sum\limits_{k=1}^{K}\alpha_{c,f}^{(\tilde{l})}[l]P_{f,k}[l]h_{f,k}[l] + N_0}\right) \tag{39}$$

$$- \frac{\left(\sum\limits_{c=1,c\neq m}^{M}\sum\limits_{f=1}^{J}\sum\limits_{k=1}^{K}P_{f,k}[l]h_{f,k}[l]\right) \times P_{j,k}[l]h_{j,k}[l]\log_2(e) \times \left(\alpha_{c,f}[l] - \alpha_{c,f}^{(\tilde{l})}[l]\right)}{\left(\sum\limits_{b=k}^{K}P_{j,b}[l]h_{j,b}[l] + \sum\limits_{c=1,c\neq m}^{M}\sum\limits_{f=1}^{J}\sum\limits_{k=1}^{K}\alpha_{c,f}^{(\tilde{l})}[l]P_{f,k}[l]h_{f,k}[l] + N_0\right)\left(\sum\limits_{b=k}^{K}P_{j,b}[l]h_{j,b}[l] + \sum\limits_{c=1,c\neq m}^{M}\sum\limits_{f=1}^{J}\sum\limits_{k=1}^{K}\alpha_{c,f}^{(\tilde{l})}[l]P_{f,k}[l]h_{f,k}[l] + N_0\right)}$$

Now, (34) has been transformed into a convex optimization problem (40) as follows

$$\max_{\mathbf{A},\{\lambda_{j,k}[l]\}} \quad \frac{1}{L}\sum_{l=1}^{L}\sum_{k=1}^{K}R_{j,k}[l] \tag{40a}$$

$$\text{s.t.} \quad \frac{1}{L}\sum_{l=1}^{L}\sum_{k=1}^{K}\sum_{m=1}^{M}\delta_1[l]\frac{\theta_{j,m}^{\tilde{lb}}[l] - \left(\alpha_{j,m}^2[l] + \lambda_{j,k}^2[l]\right)}{2} \geq \lambda, \forall j \tag{40b}$$

$$R_{j,k}^{\tilde{lb}}[l] \geq \lambda_{j,k}[l], \forall j, k, m \tag{40c}$$

$$(14), (15) \text{ and } (35d) \tag{40d}$$

### 3.3. Transmit Power Optimization

Given $\delta_1$, $\delta_0$, $\mathbf{A}$ and $\mathbf{Q}$, the optimization problem with respect to the LGU transmit power $\mathbf{P}$ can be described as follows

$$\max_{\mathbf{P}} \quad \frac{1}{L}\sum_{l=1}^{L}\sum_{k=1}^{K}R_{j,k}[l] \tag{41a}$$

$$\text{s.t.} \quad \sum_{k=1}^{K}\sum_{m=1}^{M}\alpha_{j,m}[l]\delta_1[l]\log_2\left(1 + \frac{P_{j,k}[l]h_{j,k}[l]}{\sum\limits_{b=k+1}^{K}P_{j,b}[l]h_{j,b}[l] + \sum\limits_{c=1,c\neq m}^{M}\sum\limits_{f=1}^{J}\sum\limits_{k=1}^{K}\alpha_{c,f}[l]P_{f,k}[l]h_{f,k}[l] + N_0}\right) \tag{41b}$$

$$\geq \lambda$$

$$(21) \tag{41c}$$

$$P_{j,k}[l] \geq 0, \forall j, k, l \tag{41d}$$

To facilitate derivations, the logarithmic function in (41b) can be written as the difference between two concave functions with respect to $P_{j,b}[l]$, as follows

$$R_{j,k}[l] = \log_2\left(\sum_{b=k}^{K}P_{j,b}[l]h_{j,b}[l] + \sum_{c=1,c\neq m}^{M}\sum_{f=1}^{J}\sum_{k=1}^{K}\alpha_{c,f}[l]P_{f,k}[l]h_{f,k}[l] + N_0\right) - \tilde{R}_{f,m}[l] \tag{42}$$

where

$$\tilde{R}_{f,m}[l] = \log_2\left(\sum_{b=k+1}^{K} P_{j,b}[l]h_{j,b}[l] + \sum_{c=1,c\neq m}^{M}\sum_{f=1}^{J}\sum_{k=1}^{K}\alpha_{c,f}[l]P_{f,k}[l]h_{f,k}[l] + N_0\right) \quad (43)$$

To further deal with (41b), in each iteration, we can replace $\widetilde{R}_{f,m}[l]$ with its first-order Taylor series expansion to convert (43) into a convex constraint with respect to $P_{j,b}[l]$. For $P_{j,b}^{(\tilde{l})}[l]$ in the $\tilde{l}$-th iteration, we have the following derivations

$$\tilde{R}_{f,m}[l] \leq \tilde{R}_{f,m}^{\mathrm{ub}}[l] = \log_2\left(\sum_{b=k+1}^{K} P_{j,b}^{(\tilde{l})}[l]h_{j,b}[l] + \sum_{c=1,c\neq m}^{M}\sum_{f=1}^{J}\sum_{k=1}^{K}\alpha_{c,f}[l]P_{f,k}^{(\tilde{l})}[l]h_{f,k}[l] + N_0\right)$$
$$+ B_{j,m}[l]\left(P_{j,b}[l] - P_{j,b}^{(\tilde{l})}[l]\right) \quad (44)$$

where

$$B_{j,m}[l] =$$
$$\sum_{c=1,c\neq m}^{M}\sum_{f=1}^{J}\sum_{k=1}^{K}\frac{\alpha_{c,f}[l]h_{f,k}[l]\log_2(e)}{\sum\limits_{g=1,g\neq m}^{M}\sum\limits_{f=1}^{J}\sum\limits_{k=1}^{K}\alpha_{g,\tilde{f}}[l]P_{\tilde{f},\tilde{k}}^{(\tilde{l})}[l]h_{\tilde{f},\tilde{k}}[l] + \sum\limits_{\tilde{b}=\tilde{k}+1}^{K}P_{\tilde{f},\tilde{b}}^{(\tilde{l})}[l]h_{\tilde{f},\tilde{b}}[l] + N_0} \quad (45)$$

We can obtain $\widetilde{R}_{f,m}^{\mathrm{ub}}[l]$ as the upper bound of $\widetilde{R}_{f,m}[l]$ in the Taylor series expansion. We assume that the warden can always obtain the minimum value of $\xi$ by utilizing the optimized $\tau$. To this end, it is necessary to first derive the optimal solution $\tau$ that can minimize $\xi$. Considering the noise uncertainty at the warden, $\xi$ can be expressed as

$$\xi = \mathbb{P}_{\mathrm{FA}} + \mathbb{P}_{\mathrm{MD}} = \left(\sigma_w^2 > \tau\right) + \left(P_{j,k}[l]h_{\mathrm{gw}}[l] + \sigma_w^2 < \tau\right)$$
$$= 1 - \left(\tau - P_{j,k}[l]h_{\mathrm{gw}}[l] < \sigma_w^2 < \tau\right) \quad (46)$$
$$= 1 - \int_{\max\left(\tau - P_{j,k}[l]h_{\mathrm{gw}}[l], \frac{\hat{\sigma}^2}{\rho}\right)}^{\tau} f_{\sigma_w^2}(x)dx$$

where $f_{\sigma_w^2}(x)$ is given in (20). $\tau$ is the detection threshold at the $l$-th time slot.

The first derivative of $\xi$ with respect to $\tau$ is given by

$$\frac{\mathrm{d}\xi}{\mathrm{d}\tau} = \begin{cases} -\frac{1}{2\ln(\rho)} \times \frac{1}{\tau}, & \tau \leq P_{j,k}[l]h_{\mathrm{gw}}[l] + \frac{\hat{\sigma}^2}{\rho} \\ -\frac{1}{2\ln(\rho)}\left(\frac{1}{\tau} - \frac{1}{\tau - P_{j,k}[l]h_{\mathrm{gw}}[l]}\right), & \tau > P_{j,k}[l]h_{\mathrm{gw}}[l] + \frac{\hat{\sigma}^2}{\rho} \end{cases} \quad (47)$$

When $\tau \leq P_{j,k}[l]h_{\mathrm{gw}}[l] + \hat{\sigma}^2/\rho$, $\xi$ is a decreasing function. When $\tau > P_{j,k}[l]h_{\mathrm{gw}}[l] + \hat{\sigma}^2/\rho$, $\xi$ is an increasing function. Therefore, we have

$$\tau_{\mathrm{min}} = P_{j,k}[l]h_{\mathrm{gw}}[l] + \frac{\hat{\sigma}^2}{\rho} \quad (48)$$

Substituting (48) into (46), then we have

$$\xi_{\mathrm{min}} = 1 - \frac{1}{2\ln(\rho)}\ln\left(1 + \frac{\rho P_{j,k}[l]h_{\mathrm{gw}}[l]}{\hat{\sigma}^2}\right) \quad (49)$$

Substituting (49) into problem (41c), the original problem can be expressed as

$$\ln\left(1 + \frac{\rho P_{j,k}[l]h_{\mathrm{gw}}[l]}{\hat{\sigma}^2}\right) \leq 2\varepsilon\ln\rho \quad (50)$$

Note that (50) is a concave function with respect to $P_{j,k}[l]$. It is known that the first-order approximation of any concave function is its upper bound. Therefore, for any given feasibility point $\tilde{P}_{j,k}[l], \forall l$, we have

$$\ln\left(1 + \frac{\rho P_{j,k}[l]h_{\mathrm{gw}}[l]}{\hat{\sigma}^2}\right) \leq g_1\left(P_{j,k}[l], \tilde{P}_{j,k}[l]\right), \forall l \tag{51}$$

where

$$g_1\left(P_{j,k}[l], \tilde{P}_{j,k}[l]\right) = \ln\left(1 + \frac{\rho \tilde{P}_{j,k}[l]h_{\mathrm{gw}}[l]}{\hat{\sigma}^2}\right) + \frac{\rho h_{\mathrm{gw}}[l]\left(P_{j,k}[l] - \tilde{P}_{j,k}[l]\right)}{\hat{\sigma}^2 + \rho \tilde{P}_{j,k}[l]h_{\mathrm{gw}}[l]}, \forall j, k, l \tag{52}$$

Note that any given feasibility point $\tilde{P}_{j,k}[l], \forall l$, $g_1(P_{j,k}[l], \tilde{P}_{j,k}[l])$ is a linear function with respect to transmit power $P_{j,k}[l]$. Thus, $g_1(P_{j,k}[l], \tilde{P}_{j,k}[l])$ is a convex function.

Hence, (41) can be rearranged as follows

$$\max_{\mathbf{P}} \quad \frac{1}{L}\sum_{l=1}^{L}\sum_{k=1}^{K} R_{j,k}[l] \tag{53a}$$

$$\text{s.t.} \quad \sum_{k=1}^{K}\sum_{m=1}^{M}\alpha_{j,m}[l]\delta_1[l]\left(\log_2\left(\sum_{b=k}^{K}P_{j,b}[l]h_{j,b}[l] + \sum_{c=1,c\neq m}^{M}\sum_{f=1}^{J}\sum_{k=1}^{K}\alpha_{c,f}[l]P_{f,k}[l]h_{f,k}[l] + N_0\right)\right) \\ - \tilde{R}^{ub}_{f,m}[l] \geq \lambda, \forall j \tag{53b}$$

$$g_1\left(P_{j,k}[l], \tilde{P}_{j,k}[l]\right) \leq 2\varepsilon \ln\rho \tag{53c}$$

$$P_{j,k}[l] \geq 0, \forall j, k, l \tag{53d}$$

### 3.4. UAV Trajectory Optimization

Given $\delta_1$, $\delta_0$, $\mathbf{A}$ and $\mathbf{P}$, we introduce a slack variable $s_1$, and then the optimization problem with respect to multi-UAV trajectory $\mathbf{Q}$ can be formulated as follows

$$\max_{\mathbf{Q}} \quad \frac{1}{L}\sum_{l=1}^{L} s_1 \tag{54a}$$

$$\text{s.t.} \quad s_1 \leq \sum_{k=1}^{K} R_{j,k}[l] \tag{54b}$$

$$s_1 \geq \lambda \tag{54c}$$

$$(1), (2), (3), (21) \tag{54d}$$

Since constraint (54b) is non-convex with respect to $\mathbf{Q}$, it is difficult to directly solve (54). The approximate solution of problem (54) can be obtained by adopting the successive convex approximation technique. The following Theorem 1 transforms (54) into a convex problem. Before Theorem 1 is given, the non-convex constraint (54b) is converted into a convex one using Lemma 2 below.

**Lemma 2.** *We can transform constraint (54b) into the following convex form*

$$s_1 \leq \tilde{R}^{\mathrm{lb}} \tag{55}$$

*where*

$$\tilde{R}^{\mathrm{lb}} = \sum_{m=1}^{M}\sum_{k=1}^{K}\alpha_{j,m}[l]\delta_1[l]\log_2\left(1 + \frac{\beta_{j,k}[l]}{H^2 + ||\mathbf{q}_m^{(I)}[l] - \mathbf{W}_{j,k}||^2}\right) \\ - \frac{1}{\ln 2}\sum_{m=1}^{M}\sum_{k=1}^{K}\frac{\alpha_{j,m}[l]\delta_1[l]\beta_{j,k}[l]\left(||\mathbf{q}_m[l] - \mathbf{W}_{j,k}||^2 - ||\mathbf{q}_m^{(I)}[l] - \mathbf{W}_{j,k}||^2\right)}{\left(H^2 + ||\mathbf{q}_m^{(I)}[l] - \mathbf{W}_{j,k}||^2\right)^2 + \beta_{j,k}[l]\left(H^2 + ||\mathbf{q}_m^{(I)}[l] - \mathbf{W}_{j,k}||^2\right)} \tag{56}$$

**Proof.** The detailed proof is presented in Appendix B.

Similar to the method of (41c), (21) can be transformed as follows

$$\ln\left(1 + \frac{\rho P_m[l]\beta_0}{\hat{\sigma}^2(H^2 + \|\mathbf{q}_m[l] - \mathbf{q}_w\|^2)}\right) \leq 2\varepsilon \ln \rho \tag{57}$$

Note that (57) is convex with respect to $\|\mathbf{q}_m[l] - \mathbf{q}_w\|^2$; thus, (54 g) is a convex function. In particular, we have

$$\|\mathbf{q}_m[l] - \mathbf{q}_u[l]\|^2 \geq -\|\mathbf{q}_m^{(\tilde{I})}[l] - \mathbf{q}_u^{(\tilde{I})}[l]\|^2 + 2\left(\mathbf{q}_m^{(\tilde{I})}[l] - \mathbf{q}_u^{(\tilde{I})}[l]\right)^T \times (\mathbf{q}_m[l] - \mathbf{q}_u[l]) \tag{58}$$

□

According to Lemma 2, the following Theorem 1 turns (54) into a convex constraint problem.

**Theorem 1.** *Given* $\mathbf{Q}^{(\tilde{I})} = \left\{\mathbf{q}^{(\tilde{I})}[l], \forall l\right\}$, *after* $(\tilde{I} + 1)$ *-th iterations, (54) is transformed as*

$$\max_{\mathbf{Q}} \frac{1}{L} \sum_{l=1}^{L} s_1 \tag{59a}$$

$$\text{s.t.} \quad s_1 \leq \tilde{R}^{\tilde{\text{lb}}} \tag{59b}$$

$$s_1 \geq \lambda \tag{59c}$$

$$(1), (2), (58) \tag{59d}$$

$$\ln\left(1 + \frac{\rho P_m[l]\beta_0}{\hat{\sigma}^2(H^2 + \|\mathbf{q}_m[l] - \mathbf{q}_w\|^2)}\right) \leq 2\varepsilon \ln \rho \tag{59e}$$

**Proof of Theorem 1.** According to Lemma 2, we can obtain the lower bound of $\sum\limits_{k=1}^{K} R_{j,k}[l]$ as

$$\sum_{k=1}^{K} R_{j,k}[l] > \tilde{R}^{\tilde{\text{lb}}} \tag{60}$$

As $\tilde{R}^{\tilde{\text{lb}}}$ is convex with respect to $\mathbf{Q}$, we can see that the objective function and the constraints in (59) are convex. Hence, (59) is a convex optimization problem that can be solved with the CVX tool.

The proof is completed.

□

*3.5. Proposed Algorithm*

Algorithm 1 exhibits the complete steps of the proposed iterative algorithm.

---

**Algorithm 1:** The Iterative Algorithm to Solve Problem (22).

---

1 **Initialize:** $\mathbf{A}^{(0)}, \delta_0^{(0)}\left(\delta_1^{(0)}\right), \mathbf{P}^{(0)}, \mathbf{Q}^{(0)}$ and $\tilde{l} = 0$.

2 **Repeat**

3    Solve problem (23) with given $\mathbf{A}^{(\tilde{l})}, \mathbf{P}^{(\tilde{l})}$ and $\mathbf{Q}^{(\tilde{l})}$, and then denote the solution
   as $\delta_0^{(\tilde{l}+1)}\left(\delta_1^{(\tilde{l}+1)}\right)$.

4    Solve problem (35) with given $\delta_0^{(\tilde{l}+1)}\left(\delta_1^{(\tilde{l}+1)}\right), \mathbf{P}^{(\tilde{l})}$ and $\mathbf{Q}^{(\tilde{l})}$, and then denote
   the solution as $\mathbf{A}^{(\tilde{l}+1)}$.

5    Solve problem (42) with given $\delta_0^{(\tilde{l}+1)}\left(\delta_1^{(\tilde{l}+1)}\right), \mathbf{A}^{(\tilde{l}+1)}$ and $\mathbf{Q}^{(\tilde{l})}$, and then
   denote the solution as $\mathbf{P}^{(\tilde{l}+1)}$.

6    Solve problem (55) with given $\delta_0^{(\tilde{l}+1)}\left(\delta_1^{(\tilde{l}+1)}\right), \mathbf{A}^{(\tilde{l}+1)}$ and $\mathbf{P}^{(\tilde{l}+1)}$, and then
   denote the solution as $\mathbf{Q}^{(\tilde{l}+1)}$.

7    Update $\tilde{l} = \tilde{l} + 1$.

8 **Until:** The increment of the objective value is below a predefined threshold.

---

## 4. Numerical Results and Discussion

The simulation results are presented in this section to verify the effectiveness of the proposed joint optimization scheme. The simulation parameters are as follows. A multi-UAV-assisted covert communication system consisting of $M = 2$ UAVs, $G = 12$ LGUs and a warden is considered. All LGUs are randomly distributed in an area of $1000 \times 1000$ m². LGUs are divided into $J = 6$ clusters, and each cluster contains $K = 2$ LGUs.

Suppose that the UAVs fly at a fixed altitude $H = 100$ m with a flight cycle $T = 60$ s. The number of timeslots is $N = 60$, and thus each timeslot lasts $\delta = 1$ s. The maximum flight speed of the UAVs is $V_{\max} = 40$ m/s. The noise power, channel power gain and energy conversion efficiency are, respectively, set as $N_0 = -120$ dBm, $\beta_0 = -20$ dB and $\mu = 0.85$. The maximum and minimum thresholds of $\delta_0[l]$ are rendered as $\delta_0[l]_{\max} = 0.9999$ and $\delta_0[l]_{\min} = 0.0001$ when we utilize the bisection method to search for the optimal downlink sub-slot [12].

The remaining parameters are set as $\rho_{\mathrm{dB}} = 3$, $\varepsilon = 0.005$ and $\hat{\sigma}^2 = -120$, respectively. For simplicity, the initial trajectories of UAV1 and UAV2 are designed as circles [37]. Specifically, we first calculate the geometric center of the horizontal ordinates of all LGUs as $\mathbf{L}_d = \sum_{j=1}^{J} \sum_{k=1}^{K} W_{j,k} / (J \times K) = \left[L_x, L_y\right]^{\mathrm{T}}$ to determine the half of the distance between $\mathbf{L}_d$ and the farthest LGU as $r_d = \max \left\| \mathbf{W}_{j,k} - \mathbf{L}_d \right\| / 2, \forall j, k$. Then, we set the centers of the initial trajectories as $O_1 = \left[L_x - r_d, L_y\right]^{\mathrm{T}}$ and $O_2 = \left[L_x + r_d, L_y\right]^{\mathrm{T}}$ and the radius as $r_1 = r_2 = \min([r_d/2], [V_{\max}T/2\pi])$. Hence, the position of the UAV $m$ in timeslot $l$ is given by

$$\mathbf{q}_m^{(0)}[l] = \left[x_m + r_m \sin \frac{2\pi(l-1)}{L-1}, y_m + r_m \cos \frac{2\pi(l-1)}{L-1}\right]^{\mathrm{T}} \tag{61}$$

Figure 2 shows the optimized multi-UAV trajectory for different warden positions. Here, two representative warden positions are selected to verify the effectiveness of the proposed algorithm. As expected, it can be seen that the UAVs fly as close to the LGUs as possible while keeping away from the warden by adjusting their respective trajectories. Specifically, UAV1 flies faster near the warden but slower near the LGUs mainly because this contributes to maintaining better channel conditions and improving the average uplink covert rate of LGUs.

The trajectory trend of UAV2 is similar to that of UAV1. It can be seen from Figure 2a,b that UAV1 and UAV2 cannot approach LGUs that are far apart from geometric center $L_d$ due to the limitations of the flight cycle and maximum flight speed. Instead, they relinquish

some distant LGUs to achieve the optimization objective. In fact, UAV1 and UAV2 prefer to serve LGUs with better channel conditions to improve the average uplink covert rate.

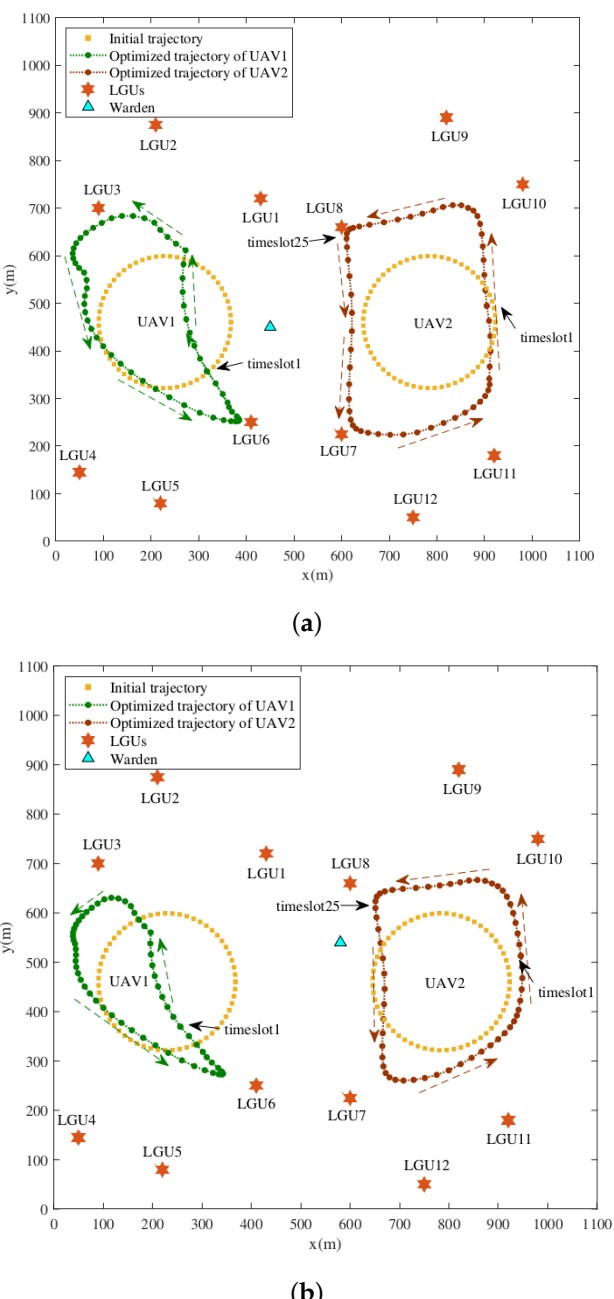

**Figure 2.** Optimized multi-UAV trajectory for different warden positions. (**a**) The warden is located at $[450, 450]^T$. (**b**) The warden is located at $[580, 540]^T$.

Figure 3 shows the UAV flight speed when the warden is located at $[450, 450]^T$. It can be seen that UAV1 has a higher flight speed in the first eight timeslots, while UAV2 has a higher flight speed in timeslots 25–34. Combined with Figure 2a, we found that UAV1 and UAV2 fly away from the warden in the same timeslots, which can greatly reduce the correctness of the warden's decision. In timeslots 20–28 and 48–54, UAV1 maintains a low flight speed because it needs to hover above the LGUs it serves for a period to receive more information. In the remaining time, UAV1 must fly to the next LGU as soon as possible at a higher speed. The speed trend of UAV2 is similar to that of UAV1.

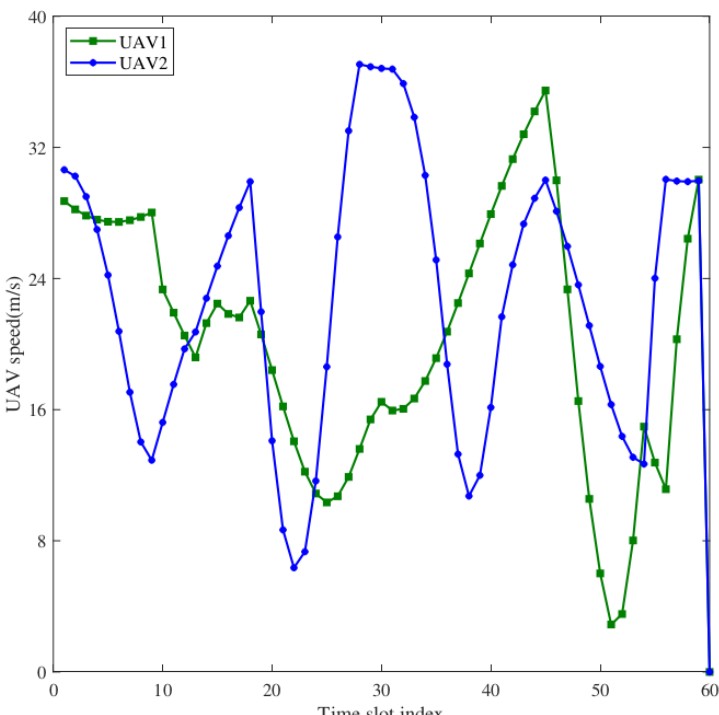

**Figure 3.** The UAV flight speed when the warden is located at $[450, 450]^{\text{T}}$.

Figure 4 shows the optimized multi-UAV trajectory for different coverage constraints. Note that the distance between the UAVs and the warden decreases when $\varepsilon$ decreases gradually. Intuitively, we can expect that, as the coverage constraint becomes more stringent, the distance between the UAVs and the warden will become larger, which is, however, inconsistent with the above simulation results. This is mainly because, although the proposed scheme synergistically optimizes the UAV transmit power and trajectory, the transmit power has a greater impact on the effect of covert communication.

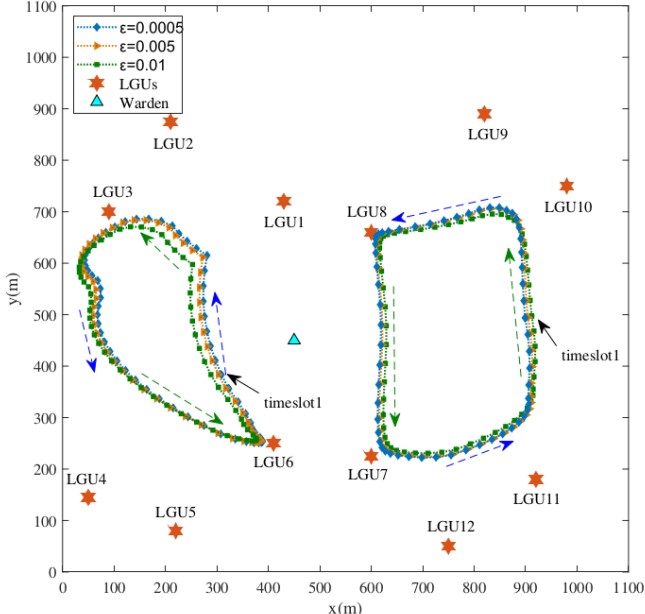

**Figure 4.** Optimized multi-UAV trajectory for different coverage constraints.

Essentially, covert communication is a technology that effectively prevents information from being intercepted, identified and located by the warden via controlling the characteristics of wireless signals. Hence, compared with $\varepsilon = 0.005$, UAV1 and UAV2 have higher transmit power when $\varepsilon = 0.01$ while flying farther away from the warden. The UAV trajectory trend for $\varepsilon = 0.0005$ is similar to those for $\varepsilon = 0.01$ and $\varepsilon = 0.005$. The simulation results in Figure 5 also confirm this interpretation.

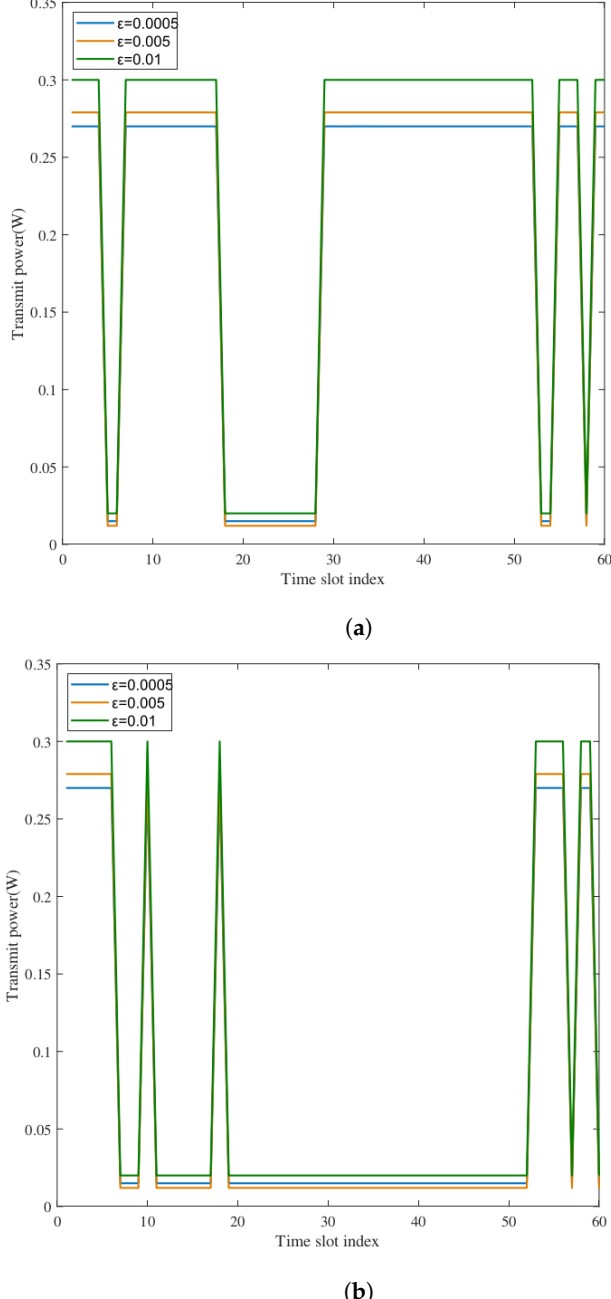

**Figure 5.** The UAVs' transmit power for different coverage constraints. (**a**) UAV1 transmit power; (**b**) UAV2 transmit power.

Figure 5 shows the multi-UAV transmit power under different coverage constraints. Note that the transmit power of UAV1 and UAV2 for $\varepsilon = 0.01$ is greater than that for $\varepsilon = 0.005$ and $\varepsilon = 0.0005$, which allows the UAVs to transmit with a higher power, thus, obtaining a higher average uplink covert rate. It can be seen that UAV1 has lower transmit power in timeslots 4–6, while UAV2 has lower transmit power in timeslots 25–35. Combined

with Figure 4, it can be found that UAV1 and UAV2 fly away from the warden with low transmit power to ensure the covertness constraint in the same timeslots. In addition, the transmit power of UAV1 and UAV2 presents an opposite variation trend because the UAVs have to adjust their respective transmit power to relieve the co-channel interference.

Figure 6 presents the average uplink covert rate versus noise uncertainty for different coverage constraints. It can be seen from Figure 6 that the average uplink covert rate improves with the increase in noise uncertainty, which is because a greater noise uncertainty will increase the difficulty for the warden to make correct decisions when detecting the transmission behavior between the UAVs and LGUs. Therefore, the increase in noise uncertainty can not only enhance information transmission between the UAVs and LGUs but also satisfy more strict covertness constraints. In addition, Figure 6 also confirms that a more loose coverage constraint helps to improve the average uplink covert rate.

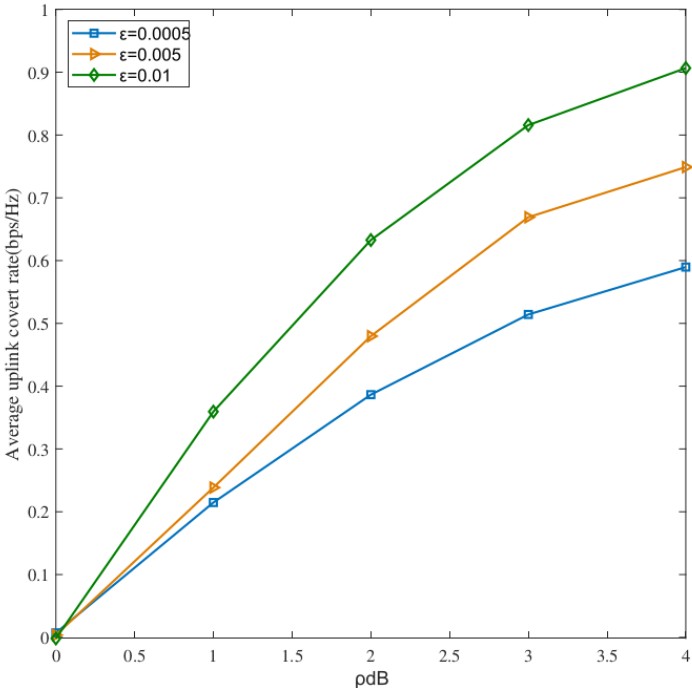

**Figure 6.** The average uplink covert rate under different noise uncertainties.

Figure 7 shows the average uplink covert rates of all LGUs under different schemes. Five schemes were selected for comparison in the simulation. Scheme 1: The initial cluster scheduling variable, sub-slot duration, LGU transmit power and multi-UAV trajectories. Scheme 2: Only the sub-slot duration, LGU transmit power and multi-UAV trajectories are optimized. Scheme 3: Only the cluster scheduling variable, sub-slot duration and multi-UAV trajectories are optimized. Scheme 4: Only the cluster scheduling variable, LGU transmit power and multi-UAV trajectories are optimized. Scheme 5: Synergistically optimize the cluster scheduling variable, sub-slot duration, LGU transmit power and multi-UAV trajectory, i.e., the proposed joint optimization scheme.

It can be seen from Figure 7 that the average uplink covert rate of all LGUs for the proposed scheme is significantly higher than that of other schemes. It is observed that the average uplink covert rate of the five schemes is enhanced with the increase in the flight cycle, which is because a large flight cycle can provide more time to the UAVs to hover above each LGU and receive more information, thereby, greatly improving the covert transmission rate. In addition, it is noted that, compared with the cluster scheduling variable and sub-slot duration optimization, the transmit power optimization of LGUs has a greater impact on the average uplink covert rate.

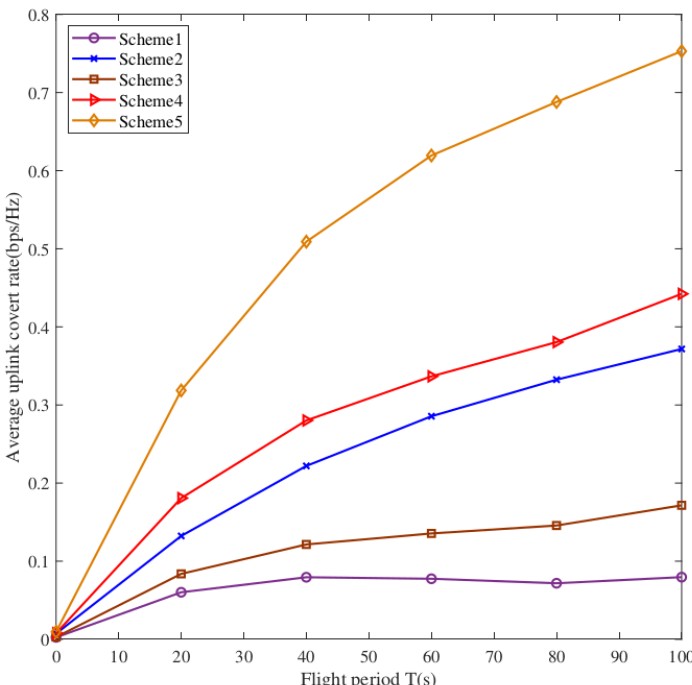

**Figure 7.** The average uplink covert rate for different optimization schemes.

Figure 8 shows the average uplink covert rate of all LGUs under different multiple access schemes. Here, the C-NOMA adopted in this paper is compared with OFDMA. As a typical OMA scheme, OFDMA divides the transmission bandwidth into several orthogonal non-overlapping subbands so that each LGU has an exclusive subband. As shown in Figure 8, the C-NOMA scheme is superior to OFDMA under the same conditions because intra-cluster NOMA can boost the spectral efficiency and elevate the average uplink covert rate.

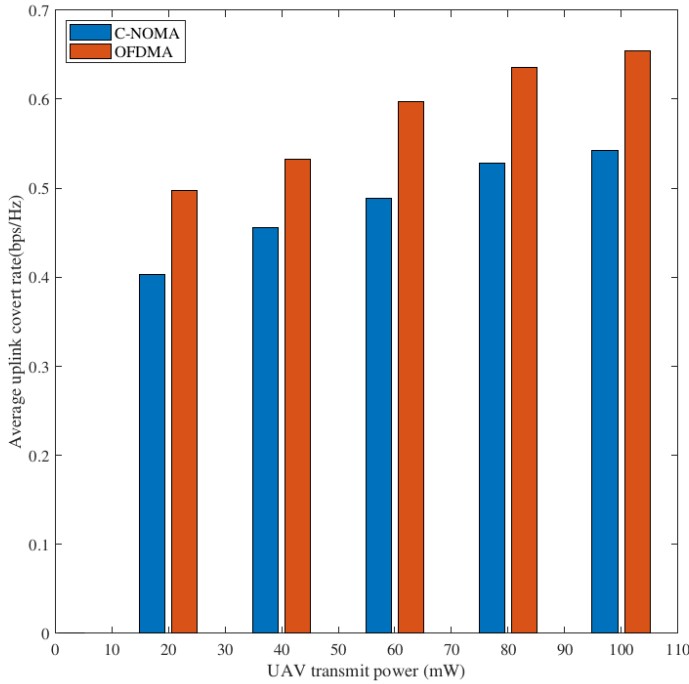

**Figure 8.** The average uplink covert rates for C-NOMA and OFDMA.

## 5. Conclusions

Aiming at secure communication in multi-user scenarios, in this paper, we proposed a multi-UAV-assisted covert communication system based on WPC and C-NOMA. Specifically, UAVs exploited WPC to transmit energy and data with LGUs in the presence of a warden. NOMA was applied to intra-cluster LGUs, while TDMA was adopted by inter-clusters to achieve an excellent trade-off between high frequency spectral efficiency and system complexity. In addition, a joint optimization scheme of the cluster scheduling variable, sub-slot allocation, LGU transmit power and multi-UAV trajectory was proposed to maximize the average uplink covert rate.

Furthermore, an iterative algorithm was proposed to achieve the optimization objective. The simulation results validate the following conclusions. First, compared with the benchmark schemes, the proposed joint optimization scheme effectively improved the average uplink covert rate of all clusters in each timeslot. Secondly, UAVs achieved covert transmission while minimizing multi-user interference by actively adjusting the flight speed and transmit power. Furthermore, the looser covertness constraint and greater noise uncertainty helped to improve the average uplink covert rate. Finally, the proposed algorithm demonstrated good convergence and could adaptively adjust the multi-UAV trajectory according to the warden's position.

**Author Contributions:** Conceptualization, X.Q.; methodology, X.W., Y.L. and Y.Z.; software, X.W., Y.L. and Y.Z.; investigation, X.W., Y.L. and Y.Z.; writing—original draft preparation, X.Q.; visualization, X.Q.; supervision, X.W. and M.X.; project administration, X.W. and M.X.; funding acquisition, X.W. All authors have read and agreed to the published version of the manuscript.

**Funding:** This research received no external funding.

**Data Availability Statement:** Not applicable.

**Conflicts of Interest:** The authors declare no conflict of interest.

## Appendix A

**Proof of Lemma 1.** Substituting (12) into (16), $\sum\limits_{k=1}^{K} R_{j,k}[l]$ can be simplified as

$$\sum_{k=1}^{K} R_{j,k}[l]$$

$$= \sum_{m=1}^{M} \sum_{k=1}^{K} \alpha_{j,m}[l]\delta_1[l]\log_2\left(1 + \frac{\frac{\mu h_{j,k}^2[l]P_m[l]\delta_0[l]}{\delta_1[l]}}{\sum_{b=k+1}^{K} \frac{\mu h_{j,b}^2[l]P_m[l]\delta_0[l]}{\delta_1[l]} + \sum_{c=1,c\neq m}^{M} \sum_{f=1}^{J} \sum_{k=1}^{K} \alpha_{c,f}[l]\frac{\mu h_{f,k}^2[l]P_m[l]\delta_0[l]}{\delta_1[l]} + N_0}\right)$$

$$= \sum_{m=1}^{M} \sum_{k=1}^{K} \alpha_{j,m}[l]\delta_1[l]\log_2\left(1 + \frac{\frac{a_{j,k}[l]\delta_0[l]}{\delta_1[l]}}{\sum_{b=k+1}^{K} \frac{a_{j,b}[l]\delta_0[l]}{\delta_1[l]} + \sum_{c=1,c\neq m}^{M} \sum_{f=1}^{J} \sum_{k=1}^{K} \alpha_{c,f}[l]\frac{a_{f,k}[l]\delta_0[l]}{\delta_1[l]} + 1}\right)$$

$$= \sum_{m=1}^{M} \sum_{k=1}^{K} \alpha_{j,m}[l]\delta_1[l]\log_2\left(\frac{\sum_{b=k}^{K} \frac{a_{j,b}[l]\delta_0[l]}{\delta_1[l]} + \sum_{c=1,c\neq m}^{M} \sum_{f=1}^{J} \sum_{k=1}^{K} \alpha_{c,f}[l]\frac{a_{f,k}[l]\delta_0[l]}{\delta_1[l]} + 1}{\sum_{b=k+1}^{K} \frac{a_{j,b}[l]\delta_0[l]}{\delta_1[l]} + \sum_{c=1,c\neq m}^{M} \sum_{f=1}^{J} \sum_{k=1}^{K} \alpha_{c,f}[l]\frac{a_{f,k}[l]\delta_0[l]}{\delta_1[l]} + 1}\right)$$

$$= \sum_{m=1}^{M} \sum_{k=1}^{K} \alpha_{j,m}[l]\delta_1[l]\log_2\left(\frac{\sum_{b=1}^{K} \frac{a_{j,b}[l]\delta_0[l]}{\delta_1[l]} + \sum_{c=1,c\neq m}^{M} \sum_{f=1}^{J} \sum_{k=1}^{K} \alpha_{c,f}[l]\frac{a_{f,k}[l]\delta_0[l]}{\delta_1[l]} + 1}{\sum_{b=2}^{K} \frac{a_{j,b}[l]\delta_0[l]}{\delta_1[l]} + \sum_{c=1,c\neq m}^{M} \sum_{f=1}^{J} \sum_{k=1}^{K} \alpha_{c,f}[l]\frac{a_{f,k}[l]\delta_0[l]}{\delta_1[l]} + 1}\right.$$

$$\cdot \frac{\sum_{b=2}^{K} \frac{a_{j,b}[l]\delta_0[l]}{\delta_1[l]} + \sum_{c=1,c\neq m}^{M} \sum_{f=1}^{J} \sum_{k=1}^{K} \alpha_{c,f}[l]\frac{a_{f,k}[l]\delta_0[l]}{\delta_1[l]} + 1}{\sum_{b=3}^{K} \frac{a_{j,b}[l]\delta_0[l]}{\delta_1[l]} + \sum_{c=1,c\neq m}^{M} \sum_{f=1}^{J} \sum_{k=1}^{K} \alpha_{c,f}[l]\frac{a_{f,k}[l]\delta_0[l]}{\delta_1[l]} + 1}$$

$$\cdots \frac{\sum_{b=k-1}^{K} \frac{a_{j,b}[l]\delta_0[l]}{\delta_1[l]} + \sum_{c=1,c\neq m}^{M} \sum_{f=1}^{J} \sum_{k=1}^{K} \alpha_{c,f}[l]\frac{a_{f,k}[l]\delta_0[l]}{\delta_1[l]} + 1}{\frac{a_{j,b}[l]\delta_0[l]}{\delta_1[l]} + \sum_{c=1,c\neq m}^{M} \sum_{f=1}^{J} \sum_{k=1}^{K} \alpha_{c,f}[l]\frac{a_{f,k}[l]\delta_0[l]}{\delta_1[l]} + 1}$$

$$\left.\times \left(\frac{a_{j,k}[l]\delta_0[l]}{\delta_1[l]} + \sum_{c=1,c\neq m}^{M} \sum_{f=1}^{J} \sum_{k=1}^{K} \alpha_{c,f}[l]\frac{a_{f,k}[l]\delta_0[l]}{\delta_1[l]} + 1\right)\right)$$

$$= \sum_{m=1}^{M} \alpha_{j,m}[l]\delta_1[l]\log_2\left(1 + \sum_{k=1}^{K} \frac{a_{j,k}[l]\delta_0[l]}{\delta_1[l]} + \sum_{c=1,c\neq m}^{M} \sum_{f=1}^{J} \sum_{k=1}^{K} \alpha_{c,f}[l]\frac{a_{f,k}[l]\delta_0[l]}{\delta_1[l]}\right) \tag{A1}$$

The proof is completed. □

## Appendix B

**Proof of Lemma 2.** According to (16), the uplink covert rate $\sum_{k=1}^{K} R_{j,k}[l]$ of the *j*-th cluster satisfies the following expression

$$\sum_{k=1}^{K} R_{j,k}[l] = \sum_{m=1}^{M} \sum_{k=1}^{K} \alpha_{j,m}[l]\delta_1[l]\log_2(1+$$

$$\left. \frac{P_{j,k}[l]h_{j,k}[l]}{\sum\limits_{b=k+1}^{K} P_{j,b}[l]\frac{\beta_0}{H^2+||\mathbf{q}_m[l]-\mathbf{W}_{j,b}||^2} + \sum\limits_{c=1,c\neq m}^{M} \sum\limits_{f=1}^{J} \sum\limits_{k=1}^{K} \alpha_{c,f}[l]P_{f,k}[l]\frac{\beta_0}{H^2+||\mathbf{q}_m[l]-\mathbf{W}_{f,k}||^2} + N_0} \right) \quad \text{(A2)}$$

$$> \sum_{m=1}^{M} \sum_{k=1}^{K} \alpha_{j,m}[l]\delta_1[l]\log_2\left(1+\frac{P_{j,k}[l]h_{j,k}[l]}{\sum\limits_{b=k+1}^{K} P_{j,b}[l]\frac{\beta_0}{H^2} + \sum\limits_{c=1,c\neq m}^{M} \sum\limits_{f=1}^{J} \sum\limits_{k=1}^{K} \alpha_{c,f}[l]P_{f,k}[l]\frac{\beta_0}{H^2} + N_0}\right) = \tilde{R}$$

Clearly, inequality $\sum\limits_{k=1}^{K} R_{j,k}[l] > \widetilde{R}$ holds.

Let $\beta_{j,k}[l] = \dfrac{P_{j,k}[l]\beta_0}{\sum\limits_{b=k+1}^{K} P_{j,b}[l]\frac{\beta_0}{H^2} + \sum\limits_{c=1,c\neq m}^{M} \sum\limits_{f=1}^{J} \sum\limits_{k=1}^{K} \alpha_{c,f}[l]P_{f,k}[l]\frac{\beta_0}{H^2} + N_0}$.

Then, $\widetilde{R}$ can be further simplified as

$$\tilde{R} = \sum_{m=1}^{M} \sum_{k=1}^{K} \alpha_{j,m}[l]\delta_1[l]\log_2\left(1+\frac{\beta_{j,k}[l]}{H^2+||\mathbf{q}_m[l]-\mathbf{W}_{j,k}||^2}\right) \quad \text{(A3)}$$

At this point, $\widetilde{R}$ is still non-convex. It is easy to observe that the lower bound of the convex function can be obtained through Taylor series expansion. Therefore, the first-order Taylor series expansion of the left side of (A3) is given as follows

$$\tilde{R} \geq A^{(\tilde{l})}[l] + B^{(\tilde{l})}[l]\left(||\mathbf{q}_m[l]-\mathbf{W}_{j,k}||^2 - ||\mathbf{q}_m^{(\tilde{l})}[l]-\mathbf{W}_{j,k}||^2\right) = \tilde{R}^{\text{lb}} \quad \text{(A4)}$$

where

$$A^{(\tilde{l})}[l] = \sum_{m=1}^{M} \sum_{k=1}^{K} \alpha_{j,m}[l]\delta_1[l]\log_2\left(1+\frac{\beta_{j,k}[l]}{H^2+||\mathbf{q}_m^{(\tilde{l})}[l]-\mathbf{W}_{j,k}||^2}\right) \quad \text{(A5)}$$

$$B^{(\tilde{l})}[l] = \frac{-\frac{1}{\ln 2}\sum\limits_{m=1}^{M} \sum\limits_{k=1}^{K} \alpha_{j,m}[l]\delta_1[l]\beta_{j,k}[l]}{\left(H^2+||\mathbf{q}_m^{(\tilde{l})}[l]-\mathbf{W}_{j,k}||^2\right)^2 + \beta_{j,k}[l]\left(H^2+||\mathbf{q}_m^{(\tilde{l})}[l]-\mathbf{W}_{j,k}||^2\right)} \quad \text{(A6)}$$

Since $\mathbf{q}_m^{(\tilde{l})}[l]$ and $\mathbf{q}_m[l]$ stand for the obtained trajectory in the $\tilde{l}$-th and $(\tilde{l}+1)$-th iterations, $A^{(\tilde{l})}[l]$ and $B^{(\tilde{l})}[l]$ are constants. The lower bound $\widetilde{R}^{\text{lb}}$ of $\widetilde{R}$ can be obtained using a Taylor series expansion, which is convex with respect the multi-UAV trajectory $\mathbf{Q}$. Thus, we can find the convex constraint (55).

The proof is completed. □

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
