# Peer review of "Multi-UAV Clustered NOMA for Covert Communications: Joint Resource Allocation and Trajectory Optimization"

_electronics, doi:10.3390/electronics11234056_

Round 1

Reviewer 1 Report

Multi-UAV Clustered NOMA for Covert Communications: Joint Resource Allocation and Trajectory Optimization is presented in the research. The article’s objective is to maximize the average uplink covert rate of all clusters in each slot by jointly optimizing the cluster scheduling variable, sub slot allocation legitimate ground users transmit power and multi unmanned aerial vehicle trajectory under thee premise of covert constraints. Overall the manuscript is well written, following are the suggestions to improve the quality of publication.

In abstract, line number 10, the sentence “Because the original problem is difficult to solve directly, this paper proposes an iterative algorithm, which divides the original problem, into four sub-problems, ……” should be re-written in more professional way.

In the manuscript the word “Figure” is referred as “Fig”, please use the journal formatting style.

Figure 1, some text is written in other than English Language, if they are symbols, description is required or translate them into English for consistency.

Too much mathematics is used in the manuscript, it is suggested that the results may be included in the article, while derivations can be provided in the supporting material.

Equation (22a, b, c… ), please use equation number for the result not with each step.

Figure sales are too congested. The scaling can be used in Figure 3 as (0, 10, 20, 30, 40, 50, 60). Same format can be used in other figures for both horizontal and vertical axis.

Reviewer 2 Report

In this study, an unmanned aerial supported communication network, a covert communication system in the multi-UAV supported Clustered-4 is examined.

This article is about maximizing the average uplink latent rate of all clusters in each socket jointly by 8. Also optimizing cluster timing variable, subspace allocation, LGUs transmit power and work with multiple UAV 9. This article proposes an iterative algorithm that directly divides the original problem into four.

He also developed an algorithm and compared the results of the algorithm.

In this sense, it will make a significant contribution to the study as an application. However, the purpose of the study should be clearly emphasized in the abstract section. The reference part of the study is sufficient. The mathematical background of the study is also sufficient. In the conclusion part of the study, the outputs of the study should be clearly emphasized. In addition, the notations that should be explained in equations 16-18 should be specified.

Reviewer 3 Report

1.      A hard space should be inserted between the numerical value and the measurement unit (lines 296÷299).

22. A hard space should be inserted between the text and literature reference number (eg. line 27 and 36).  Comment applies to the entire text of the article.

33. The editing error on line 85- Original text “… tecnnical …...". Corrected text “… technical …...".

44. Authors should add more textual description to the content of the article regarding the physical interpretation of individual elements of analytical relationships presented on pages ranging from 7 pages to 16 pages.

55. In Figure 1, the descriptions written in Chinese should be translated.
